# Pretransplant survival of patients with end-stage heart failure under competing risks

**Kevin B. Smith**[1¤a], **Tseeye Odugba Potters**[2¤b], **Gabriel Lopez Zenarosa**[1¤c]*

**1** Systems Engineering and Engineering Management, University of North Carolina at Charlotte, Charlotte, North Carolina, United States of America, **2** Health Informatics, University of North Carolina at Charlotte, Charlotte, North Carolina, United States of America

¤a Current address: Industrial and Operations Engineering, University of Michigan, Ann Arbor, Michigan, United States of America
¤b Current address: Healthgrades, Denver, Colorado, United States of America
¤c Current address: Management Science and Information Systems, Rutgers University, Newark and New Brunswick, Piscataway, New Jersey, United States of America
* zenarosa@business.rutgers.edu

**Data Availability Statement:** Data cannot be shared publicly as it is owned by the United Network for Organ Sharing. We do not have permission to distribute the data, however, the data may be requested from the Organ Procurement

## Abstract

Heart transplantation is the gold standard of care for end-stage heart failure in the United States. Donor hearts are a scarce resource, however the current allocation policy—proposed in 2016 and implemented in 2018—has not addressed certain disparities. Between 2005 and 2016, the number of active candidates increased 127%, whereas transplant rates decreased 27.8%. Pretransplant mortality rates declined steadily for that period from 14.6 to 9.7, especially for candidates with mechanical circulatory assistive devices (MCSDs). This study reports survival analyses of candidates for heart transplantation list under competing events of transplantation and MCSD implantation. We queried the transplant data for a cohort of adult patients (age $\geq$ 16) without MCSDs prior to listing for transplantation between 2005 and 2014 ($n$ = 23,373). We used cause-specific and subdistribution hazards models as multivariate regressions for all competing events. Patients listed as low priority for transplantation are less likely to require implantation but less likely to survive after 1,000 days of listing than patients listed at higher priorities. The current policy does not address this disparity as it focuses on stratifying patients with different types of MCSD. Clinical characteristics must be considered in prioritization.

## Introduction

Heart transplantation is the gold standard of care for end-stage heart failure in the United States (US). The number of active candidates waitlisted for heart transplantation increased by 127% (from 1,262 to 2,862) between 2005 and 2016, while the transplant rates (i.e., transplants per 100 waitlist years) decreased by 27.8% (from 129.0 to 93.1) over the same time period [1]. The rates of mortality (i.e., deaths per 100 waitlist years) over the same years, however, decreased as a result of increased usage of mechanical circulatory support devices (MCSDs), specifically ventricular assistive devices (VADs) [1]. In 2016, the Organ Procurement and Transplantation Network (OPTN) developed the new heart-allocation policy [2] to stratify

and Transplantation Network. To obtain the data, a Data Use Agreement must be signed and approved by OPTN. Please refer the following URL: (https://optn.transplant.hrsa.gov/data/request-data/). The authors did not have any special privileges in accessing this data that other authors would not have.

**Funding:** This work was supported by funds provided by The University of North Carolina at Charlotte (GLZ, Faculty Research Grant, https://research.charlotte.edu/departments/center-research-excellence-cre/locating-funding/internal-funding-programs), University Professional Internship Program (KBS, https://career.charlotte.edu/upip), Levine Scholars Program (KBS, https://levinescholars.charlotte.edu), and Honors College (KBS, https://honorscollege.charlotte.edu). The funders had no role in study design, data collection and analysis, decision to publish, or preparation of the manuscript.

**Competing interests:** The authors have declared that no competing interests exist.

MCSD-supported adult patients [3, 4], and it was eventually implemented in October of 2018. Tiers 1–4 of the new policy are now reserved for and based on the stratification of MCSDs. Part of the rationale for stratification is the increased use of MCSDs as a bridge-to-transplant (BTT) therapy [5], which affects the dynamics of heart allocation and transplantation. Thus, proper pretransplant survival analyses must account for competing events [6].

When considering pretransplant mortality, the event of death competes with other events of interest, such as transplantation and MCSD implantation, which alters the etiology and prognosis [5] and, ultimately, the survival of patients. Competing risks models were developed to ascertain proper mortality rates in studies where a traditional Kaplan-Meier survival estimate exhibits a strong bias [7] or where doctors cannot evaluate the potential of an event-specific intervention [8]. We are motivated to properly measure and estimate the survival probability of patients on the United Network for Organ Sharing (UNOS) waitlist under competing risks to potentially highlight systematic biases and possible areas of improvement in organ allocation and patient status classification.

This paper reports on competing-risk analyses of pretransplant survival of patients with end-stage heart failure under three events of interest: implantation, transplantation, and death. We use the transplant registry maintained by the UNOS to generate two models: the cause-specific Cox Proportional Hazards and the Fine-Gray subdistribution hazards models. We found the lowest-priority patients without MCSDs are disadvantaged under the current policy (as well as under the new policy), which prioritize patients with MCSD.

## Materials and methods

### Ethics

The Internal Review Board of the University of North Carolina at Charlotte exempted our study from review and permitted us to obtain data from UNOS and perform analyses on the September 2016 version of the UNOS Standard Transplant Analysis Research (STAR) files, which contain the heart transplantation waitlist in the US from which October 2018 policy change was based [9]. (The results from this study are robust to the October 2018 policy change, for which an accurate one-year UNOS survival analysis study [i.e., using stable data] only can be conducted from October 2021).

### Patient selection

We used 10 years of stable waitlist data between January 1, 2005 and December 31, 2014, with the end point chosen to align with the latest available report (with respect STAR files version date) from OPTN and the Scientific Registry of Transplant Recipients (SRTR) [1]. From the initial 2005–2014 cohort of 34,833 patients waitlisted for heart transplantation, we excluded 4,818 patients aged less than 16 years, 5,745 implanted with an MCSD prior to listing, and 897 having previously received a heart transplant, resulting in a final study sample size of 23,373 (Fig 1).

Priority for heart transplantation considers the recipients' distances from the donor hospital, waiting times, and medical needs [10]. The medical urgency of a recipient is measured using UNOS Statuses in the heart allocation system, which, in 2005–2014, were: Status 1A (most urgent), Status 1B, Status 2 (least urgent), and Status 7 (inactive).

### Disease classification

The OPTN uses six main categories of heart disease from 30 primary diagnoses, which are: cardiomyopathy, coronary heart disease, congenital heart disease, valvular heart disease,

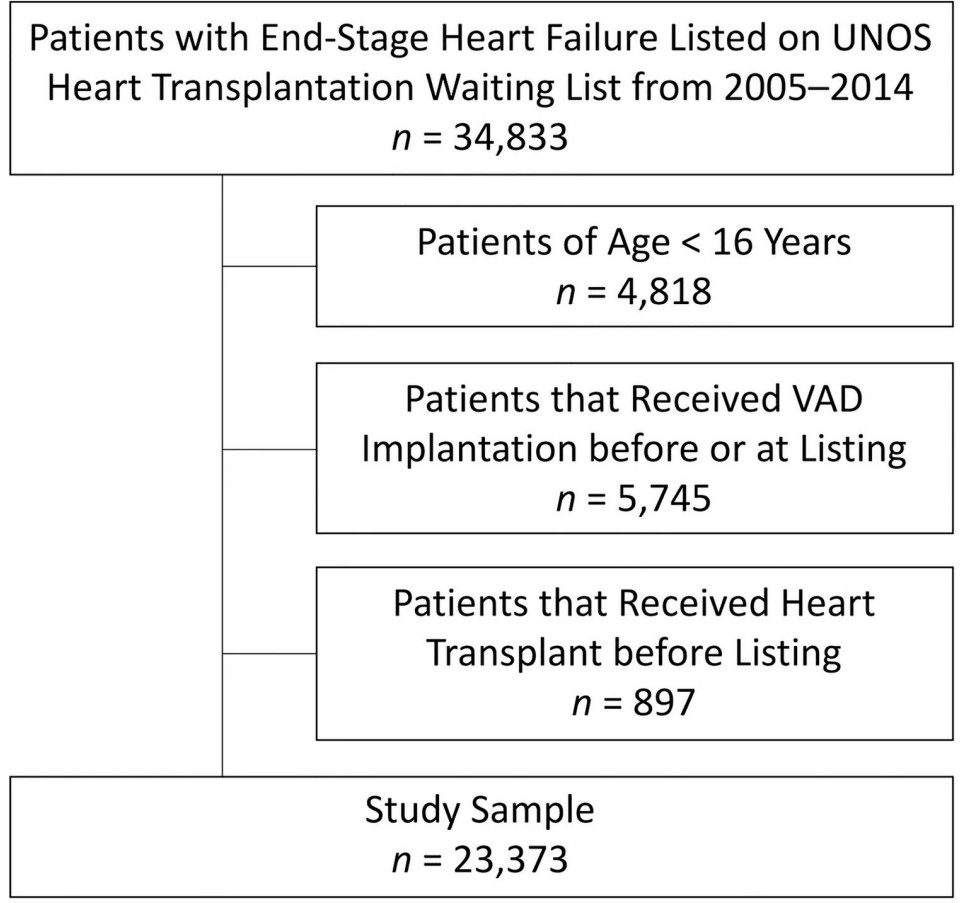

**Fig 1. Flow diagram for the exclusion of patients in the UNOS STAR files 2005 to 2014.**

retransplant graft failure, and other. We use five of these categories—excluding retransplant graft failure, which does not occur in our final cohort of patients. The category defined as other includes patients whose diagnosis at listing was not indicated ($n = 5$) or indicated as unknown ($n = 568$).

## Selection of survival prediction variables

We considered all the at-listing variables from the UNOS STAR files for heart transplantation as potential covariates for a pretransplant analysis. We used R 4.1.0 [11] to perform a mixed selection of variables available from the STAR files using the `step` function on the cause-specific hazard of death (i.e., the event of interest). From the resulting selection, we further excluded variables with $p$-values over 0.05, but we included categorical variables for which the $p$-values of the baseline categories are at or under 0.05.

## Statistical analyses

We report cumulative incidences of death overall, as well as by sex and disease classification using non-parametric cumulative incidence functions. To identify potential systemic biases in UNOS prioritization, we also report survival analyses stratified by UNOS Status under the competing events of death, transplant, and implant using non-parametric cumulative

incidence functions, cause-specific Cox Proportional Hazards model (Cox model), and Fine-Gray subdistribution hazards model (Fine-Gray model) [12]. Holistic interpretations of the last two models are available [12, 13]. We used the significant variables regressed on cause-specific hazard of death for both the Cox and Fine-Gray models.

## Results

### Patient characteristics

The study sample consisted of mainly Caucasians and male patients (Table 1). Patients in this study were diagnosed with five main categories of heart disease, over half of which were diagnosed with Cardiomyopathy at listing. The UNOS Status statistics reported were designated at listing. The baseline patient characteristics are stratified by event (Table 1) and by UNOS Status (Table 2).

### Cumulative incidence of patient survival

Cumulative incidence of death for the entire study is 5.09% (95% Confidence Interval [CI]: 4.81%-5.39%) at Year 1, 7.34% (95% CI: 6.96%-7.73%) at Year 5, and 8.71% (95% CI: 7.63%-9.94%) at Year 10 (Fig 2a). Female (Year 1: 4.29 [95% CI: 3.79%-4.84%], Year 5: 6.17 [95% CI: 5.51%-6.90%], and Year 10: 7.44 [95% CI: 5.81%-9.53%]) and male (Year 1: 5.38% [95% CI: 5.04%-5.74%], Year 5: 7.74% [95% CI: 7.30%-8.21%], and Year 10: 9.22% [95% CI: 7.81%-10.89%]) patients' cumulative incidences are shown in Fig 2b. When the cumulative incidence of death is separated by primary diagnosis, patients with Cardiomyopathy have one- and five-year cumulative incidences of 4.67% (95% CI: 4.31%-5.07%) and 6.45% (95% CI: 5.98%-6.96%), respectively, whereas those with Congenital Heart Disease have 6.47% (95% CI: 5.03%-8.30%) and 10.59% (95% CI: 8.49%-13.21%), respectively (Fig 2c).

**Table 1. Baseline patient characteristics by event.**

| Demographic Feature | | Event Free (*n* = 4,495) | Death (*n* = 1,441) | Transplant (*n* = 13,020) | Implant (*n* = 4,417) | Sample (*n* = 23,373) |
|---|---|---|---|---|---|---|
| Sex | Male | 73.6% | 78.1% | 72.5% | 76.4% | 73.8% |
| | Female | 26.4% | 21.9% | 27.5% | 23.6% | 26.2% |
| UNOS Status | 1A | 7.9% | 14.4% | 14.9% | 16.8% | 13.9% |
| | 1B | 26.1% | 34.0% | 41.6% | 47.9% | 39.4% |
| | 2 | 63.0% | 46.8% | 41.9% | 30.9% | 44.2% |
| | 7 | 2.9% | 4.8% | 1.6% | 4.5% | 2.6% |
| Ethnicity | White | 68.1% | 68.0% | 68.6% | 65.0% | 67.8% |
| | Black | 21.3% | 19.6% | 19.3% | 25.2% | 20.8% |
| | Hispanic | 7.3% | 9.1% | 8.0% | 6.4% | 7.6% |
| | Other | 3.3% | 3.3% | 4.1% | 3.4% | 3.8% |
| Age at Listing | Minimum | 16 years | 16 years | 16 years | 16 years | 16 years |
| | Median | 54 years | 56 years | 55 years | 55 years | 55 years |
| | Maximum | 76 years | 79 years | 78 years | 78 years | 79 years |
| Primary Diagnosis | Cardiomyopathy | 50.7% | 47.6% | 53.7% | 58.9% | 53.7% |
| | Coronary Heart Disease | 38.1% | 41.2% | 36.8% | 37.5% | 37.5% |
| | Congenital Heart Disease | 5.8% | 5.6% | 4.3% | 1.3% | 4.1% |
| | Valvular Heart Disease | 1.9% | 2.5% | 2.4% | 0.9% | 2.0% |
| | Other | 3.5% | 3.1% | 2.7% | 1.7% | 2.7% |

**Table 2. Baseline patient characteristics by UNOS status.**

| Demographic Feature | | UNOS Status 1A (*n* = 3,244) | UNOS Status 1B (*n* = 9,200) | UNOS Status 2 (*n* = 10,326) | UNOS Status 7 (*n* = 603) |
|---|---|---|---|---|---|
| Sex | Male | 72.7% | 73.9% | 74.0% | 73.5% |
| | Female | 27.3% | 26.1% | 26.0% | 26.5% |
| Ethnicity | White | 62.3% | 62.7% | 74.1% | 66.3% |
| | Black | 23.6% | 25.5% | 15.6% | 22.2% |
| | Hispanic | 8.9% | 8.2% | 6.6% | 8.1% |
| | Other | 5.1% | 3.5% | 3.6% | 3.3% |
| Age at Listing | Minimum | 16 years | 16 years | 16 years | 16 years |
| | Median | 53 years | 55 years | 56 years | 53 years |
| | Maximum | 79 years | 78 years | 78 years | 73 years |
| Primary Diagnosis | Cardiomyopathy | 58.3% | 58.0% | 48.4% | 52.7% |
| | Coronary Heart Disease | 33.3% | 34.8% | 41.0% | 41.0% |
| | Congenital Heart Disease | 3.5% | 2.8% | 5.6% | 2.5% |
| | Valvular Heart Disease | 2.0% | 1.9% | 2.2% | 1.5% |
| | Other | 2.9% | 2.5% | 2.8% | 2.7% |

## Predictors of survival

Table 3 lists all of the at-listing variables available from the UNOS STAR files for heart transplantation; the covariates obtained by mixed selection and tested for significance are marked (*). Table 3 also lists the hazard ratios with respect to the events of interest and the regression models (i.e., cause-specific and subdistribution hazards models). We note that the cause-specific hazard ratio (csHR) represents the rate of the event of interest in those patients that are event-free; thus, csHR provides the estimated etiological effects of the variables [14]. In contrast, the subdistribution hazard ratio (sdHR) provides the prognostic effects of the variables [14].

The interpretation of the sdHR requires some care, however [15]. We can assume that if a variable increases the subdistribution hazard, it will also increase the incidence of the event of interest, but we cannot conclude that these two are in the same magnitude. Thus, using the value of a covariate's sdHR only approximately describes the effect of that variable on the incidence of the event of interest.

When presenting the estimated HRs, we note qualitatively similar and different effects across competing events as well as the hazards models [16]. For example, UNOS Status 2 has a protective effect on the cause-specific hazards of death (csHR 0.46 [95% CI: 0.39-0.55]), transplant (csHR 0.31 [95% CI: 0.29-0.33]), and implant (csHR 0.25 [95% CI: 0.22-0.28]), whereas its effect on the subdistribution hazard of death is insignificant (sdHR 1.05 [95% CI: 0.86-1.27]). One may interpret this insignificant effect on the incidence of death as non-indicative of pretransplant death, but, as recommended elsewhere [17], we present csHRs and sdHRs to provide decision-makers the complete estimated etiological and prognostic effects [14], respectively, of the variables in our multivariate analyses. The final set of covariates is obtained through the mixed selection of variables for the cause-specific hazard of death and is used in the statistical analyses previously described.

## Predicted survival by UNOS status

For death, transplant, and implant events, the average one-year non-parametric cumulative incidences are: 5.09% (95% CI: 4.81%-5.39%), 52.26% (95% CI: 51.60%-52.93%), and 12.64% (95% CI: 12.20%-13.09%), respectively. At five years, those incidences for the same three

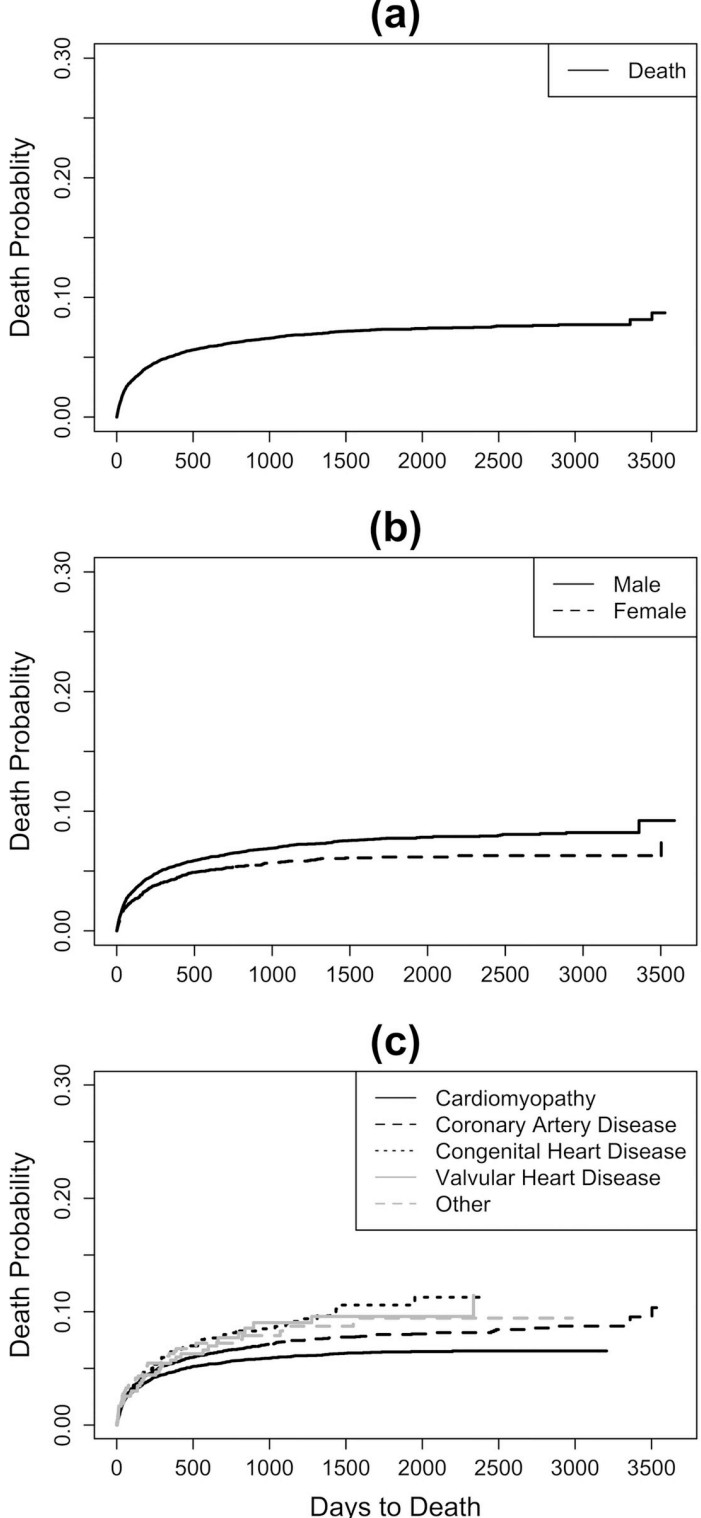

**Fig 2. Non-parametric cumulative incidence of death in 2005–2014: (a) Overall, as well as by (b) sex and (c) primary diagnosis.**

**Table 3. All UNOS STAR files at-listing variables and covariates obtained by mixed selection of variables.**

| Variable | | Cause-specific Hazards Model | | | Subdistribution Hazards Model | | |
|---|---|---|---|---|---|---|---|
| | | Death | Transplant | Implant | Death | Transplant | Implant |
| UNOS Status | 1B* | 0.65 (0.07) | 0.63 (0.02) | 0.67 (0.04) | 0.87 (0.09) | 0.72 (0.03) | 0.95 (0.05) |
| | 2* | 0.46 (0.09) | 0.31 (0.03) | 0.25 (0.06) | 1.05 (0.10) | 0.44 (0.04) | 0.65 (0.06) |
| | 7* (baseline: 1A) | 1.25 (0.11) | 0.37 (0.06) | 0.70 (0.08) | 1.63 (0.15) | 0.40 (0.08) | 1.14 (0.09) |
| Sex: Female* (baseline: Male) | | 0.80 (0.07) | 0.90 (0.02) | 0.93 (0.05) | 0.76 (0.08) | 0.93 (0.03) | 1.05 (0.05) |
| Ethnicity | White | 1.17 (0.14) | 0.93 (0.04) | 0.88 (0.09) | 0.86 (0.15) | 1.01 (0.05) | 1.09 (0.09) |
| | Black | 0.97 (0.14) | 0.88 (0.04) | 0.87 (0.09) | 0.89 (0.07) | 0.96 (0.03) | 1.05 (0.04) |
| | Hispanic (baseline = Other) | 1.29 (0.15) | 0.95 (0.05) | 0.85 (0.10) | 1.16 (0.10) | 1.01 (0.04) | 0.87 (0.06) |
| Age at Listing* | | 1.01 (0.00) | 1.01 (0.00) | 1.01 (0.00) | 1.00 (0.00) | 1.01 (0.00) | 1.00 (0.00) |
| Primary Diagnosis | Coronary Artery Disease | 0.91 (0.06) | 0.92 (0.02) | 0.86 (0.04) | 1.03 (0.07) | 0.97 (0.02) | 0.93 (0.04) |
| | Congenital Heart Disease | 0.98 (0.13) | 0.80 (0.05) | 0.37 (0.14) | 1.20 (0.14) | 0.95 (0.05) | 0.42 (0.14) |
| | Valvular Heart Disease | 1.30 (0.15) | 0.95 (0.06) | 0.49 (0.16) | 1.34 (0.17) | 1.12 (0.06) | 0.46 (0.16) |
| | Other Primary Diagnosis | 1.10 (0.14) | 0.97 (0.05) | 0.74 (0.12) | 1.12 (0.16) | 1.09 (0.05) | 0.68 (0.12) |
| (baseline: Cardiomyopathy) | | | | | | | |
| Height (cm)* | | 0.97 (0.01) | 1.01 (0.00) | 1.01 (0.01) | 0.97 (0.01) | 1.01 (0.00) | 1.02 (0.01) |
| Weight (kg)* | | 1.02 (0.01) | 0.98 (0.00) | 0.99 (0.01) | 1.03 (0.01) | 0.97 (0.00) | 0.99 (0.01) |
| Body Mass Index* | | 0.94 (0.02) | 1.03 (0.01) | 1.02 (0.02) | 0.92 (0.03) | 1.04 (0.01) | 1.05 (0.02) |
| Blood Group | AB | 1.21 (0.14) | 1.70 (0.04) | 1.27 (0.09) | 0.83 (0.06) | 1.56 (0.02) | 0.85 (0.03) |
| | B | 0.99 (0.08) | 1.00 (0.02) | 1.06 (0.05) | 0.77 (0.16) | 2.56 (0.05) | 0.60 (0.09) |
| | O (baseline: A) | 0.96 (0.05) | 0.61 (0.02) | 0.82 (0.03) | 0.78 (0.09) | 1.55 (0.03) | 0.88 (0.05) |
| Total Serum Albumin (g/dL)* | | 0.67 (0.02) | 1.45 (0.01) | 0.99 (0.01) | 0.64 (0.02) | 1.48 (0.01) | 0.79 (0.01) |
| Serum Creatinine (mg/dL)* | | 1.16 (0.01) | 1.01 (0.01) | 1.01 (0.02) | 1.15 (0.01) | 1.00 (0.01) | 0.96 (0.02) |
| No Diabetes | | 0.86 (0.15) | 1.14 (0.06) | 1.21 (0.11) | 1.27 (0.18) | 0.98 (0.07) | 0.90 (0.11) |
| Type I Diabetes | | 1.00 (0.19) | 1.28 (0.08) | 0.81 (0.16) | 1.44 (0.13) | 1.19 (0.06) | 0.60 (0.11) |
| Type II Diabetes | | 0.89 (0.15) | 1.12 (0.07) | 1.34 (0.11) | 1.04 (0.07) | 0.95 (0.02) | 1.09 (0.04) |
| Other Type Diabetes (baseline: Unknown) | | 0.82 (0.52) | 0.97 (0.19) | 1.57 (0.31) | 0.98 (0.59) | 0.99 (0.18) | 1.52 (0.27) |
| Uses Cigarettes | | 1.08 (0.06) | 0.90 (0.02) | 0.91 (0.04) | 1.22 (0.07) | 0.92 (0.02) | 0.97 (0.04) |
| Has Abstained from Cigarette Use ≥ 60 Months | | 1.01 (0.07) | 0.99 (0.02) | 1.07 (0.04) | 0.99 (0.08) | 0.96 (0.03) | 1.07 (0.04) |
| Cardiac Output (CO, L/min)* | | 0.97 (0.02) | 0.98 (0.01) | 0.99 (0.01) | 0.97 (0.02) | 0.99 (0.01) | 1.01 (0.01) |
| CO Obtained while on Vasodilaters or Inotropes (V/I) | | 0.70 (0.19) | 0.96 (0.08) | 0.73 (0.15) | 0.81 (0.22) | 1.10 (0.09) | 0.84 (0.16) |
| Unknown if CO Obtained while on V/I | | 0.96 (0.16) | 0.89 (0.06) | 0.80 (0.12) | 1.11 (0.17) | 0.98 (0.07) | 0.85 (0.11) |
| Pulmonary Artery Systolic Pressure (PASP, mmHg)* | | 1.01 (0.00) | 1.00 (0.00) | 1.00 (0.00) | 1.00 (0.00) | 1.00 (0.00) | 1.00 (0.00) |
| PASP Obtained while on V/I | | 1.51 (0.88) | 1.93 (0.29) | 3.34 (0.54) | 1.32 (0.79) | 1.22 (0.33) | 2.44 (0.58) |
| Unknown if PASP Obtained while on V/I | | 2.25 (0.66) | 1.07 (0.22) | 2.76 (0.41) | 1.34 (0.51) | 0.77 (0.21) | 2.98 (0.45) |
| Pulmonary Artery Diastolic Pressure (PADP, mmHg) | | 1.00 (0.01) | 1.00 (0.00) | 1.00 (0.00) | 1.01 (0.01) | 1.00 (0.00) | 1.00 (0.00) |
| PADP Obtained while on V/I | | 1.51 (0.89) | 0.62 (0.30) | 0.48 (0.53) | 1.45 (0.81) | 0.82 (0.33) | 0.47 (0.60) |
| Unknown if PADP Obtained while on V/I | | 0.89 (0.66) | 0.88 (0.22) | 0.49 (0.42) | 1.65 (0.52) | 1.04 (0.21) | 0.43 (0.46) |
| Pulmonary Artery Mean Pressure (PAMP, mmHg) | | 1.00 (0.01) | 0.99 (0.00) | 0.99 (0.00) | 1.00 (0.01) | 0.99 (0.00) | 1.00 (0.00) |
| PAMP Obtained while on V/I | | 0.63 (0.27) | 1.02 (0.10) | 1.15 (0.20) | 0.52 (0.27) | 0.99 (0.11) | 1.38 (0.19) |
| Unknown if PAMP Obtained while on V/I | | 0.69 (0.26) | 1.01 (0.09) | 0.82 (0.19) | 0.71 (0.28) | 1.00 (0.10) | 1.02 (0.18) |
| Pulmonary Capillary Wedge Pressure (PCWP, mmHg)* | | 1.01 (0.00) | 1.01 (0.00) | 1.01 (0.00) | 0.99 (0.01) | 1.01 (0.00) | 1.00 (0.00) |
| PCWP Obtained while on V/I* | | 1.31 (0.18) | 0.93 (0.07) | 0.76 (0.13) | 1.56 (0.21) | 1.01 (0.09) | 0.72 (0.13) |
| Unknown if PCWP Obtained while on V/I* | | 1.50 (0.18) | 1.14 (0.07) | 1.12 (0.13) | 1.22 (0.19) | 1.14 (0.08) | 0.95 (0.12) |
| On Inotropes* | | 1.39 (0.09) | 1.12 (0.03) | 0.38 (0.04) | 1.77 (0.11) | 3.42 (0.05) | 0.27 (0.05) |
| On Life Support | | 0.89 (0.09) | 0.88 (0.04) | 3.01 (0.05) | 0.66 (0.12) | 0.28 (0.06) | 4.43 (0.05) |
| On Other Mechanism of Life | | 1.19 (0.15) | 0.81 (0.06) | 0.66 (0.09) | 1.43 (0.18) | 1.22 (0.08) | 0.69 (0.09) |
| Number of Previous Non-heart Transplants* | | 1.90 (0.13) | 1.03 (0.07) | 0.93 (0.17) | 1.91 (0.12) | 0.94 (0.08) | 0.81 (0.18) |
| Had Prior Cardiac Surgeries* | | 1.23 (0.05) | 0.94 (0.02) | 1.05 (0.04) | 1.22 (0.06) | 0.93 (0.02) | 1.08 (0.03) |

(*Continued*)

**Table 3.** (Continued)

| Variable | Cause-specific Hazards Model | | | Subdistribution Hazards Model | | |
|---|---|---|---|---|---|---|
| | Death | Transplant | Implant | Death | Transplant | Implant |
| Unknown History of Prior Cardiac Surgeries* | 1.37 (0.19) | 0.88 (0.06) | 1.51 (0.12) | 1.33 (0.21) | 0.75 (0.08) | 1.39 (0.14) |
| Symptomatic of Cerebrovascular Disease* | 1.31 (0.10) | 1.15 (0.04) | 1.34 (0.07) | 1.14 (0.11) | 1.04 (0.04) | 1.14 (0.07) |
| Unknown to be Symptomatic of Cerebrovascular Disease* | 1.28 (0.20) | 1.13 (0.09) | 0.83 (0.20) | 1.57 (0.23) | 1.16 (0.10) | 0.62 (0.21) |
| Has Previous Malignancy | 0.92 (0.10) | 0.98 (0.03) | 1.04 (0.06) | 0.88 (0.11) | 1.00 (0.04) | 1.01 (0.06) |
| Unknown to Have Previous Malignancy | 1.32 (0.18) | 0.78 (0.08) | 0.47 (0.19) | 1.41 (0.22) | 0.99 (0.08) | 0.56 (0.18) |

Numeric and parenthesized numeric entries are the hazard ratios (standard errors) for each variable and respective event and model; Covariates derived via mixed-selection of variables are indicated with asterisks (*).

events are: 7.34% (95% CI: 6.96%-7.73%), 61.70% (95% CI: 61.00%-62.40%), and 24.96% (95% CI: 24.30%-25.64%), respectively (Fig 3). We observe a crossover on the cumulative incidence for the death event between patients listed at UNOS Statuses 1A and 2 at 1,002 days, 6.90% (95% CI: 6.04%-7.89%) and 6.91% (95% CI: 6.38%-7.47%), respectively (Fig 3a). We observe a

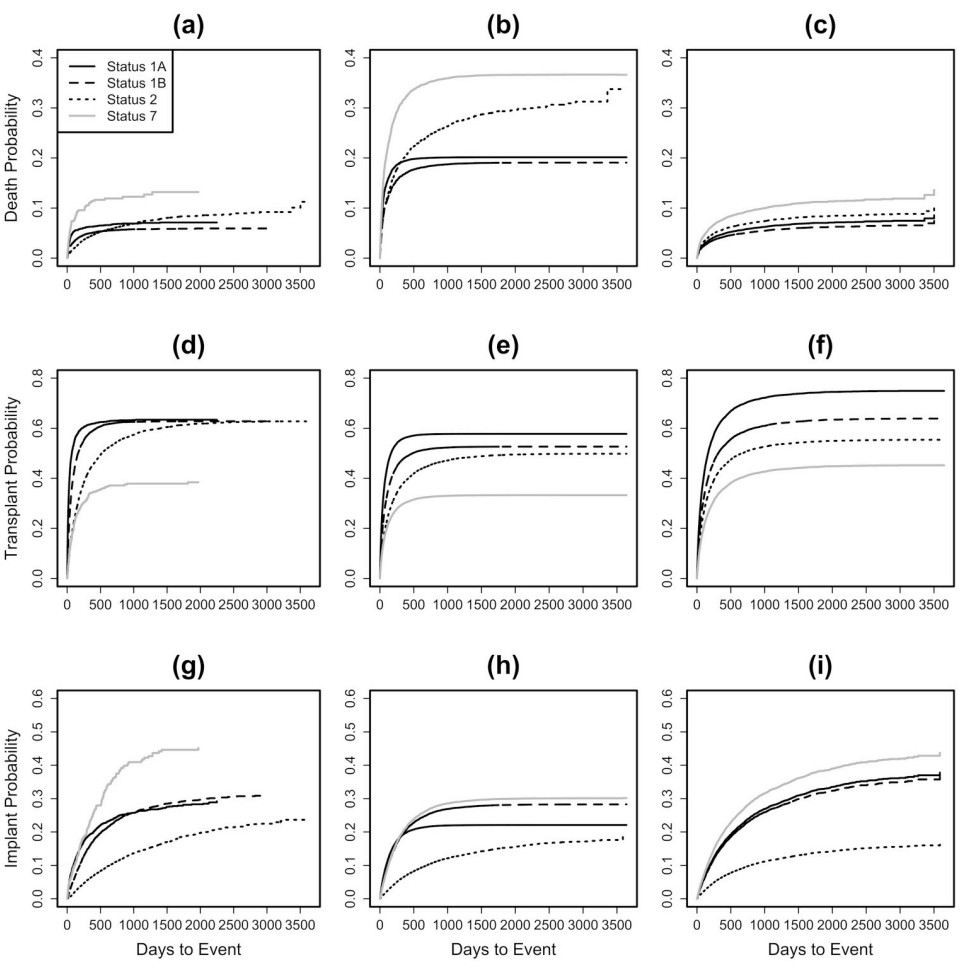

**Fig 3. Cumulative incidences of death (a–c), transplantation (d–f), and implantation (g–i) by UNOS status in 2005–2014 using three survival analysis models: Non-parametric (a, d, g), cox model (b, e, h), and Fine-Gray model (c, f, i).**

similar crossover in the Cox model for the death event between patients listed at UNOS Statuses 1A and 2 at 300 days, 19.09% (95% CI: 10.40%-26.94%) and 19.09% (95% CI: 8.38%-28.55%), respectively (Fig 3b). The Fine-Gray model, by design, does not exhibit this crossover (Fig 3c): Status 2 patients have a higher cumulative incidence rate for the death event than the UNOS Status 1A and 1B patient groups.

Transplant rates for the non-parametric cumulative incidence plot indicate that only Status 2 patients are receiving transplantation after 500 days and before 2000 days (Fig 3d). The Cox model shows a similar pattern by predicting only Status 2 patients as receiving transplantation after 600 days through around 2,000 days (Fig 3e). However, the Fine-Gray model does not illustrate this sloping difference (Fig 3f).

Implant rates, for all three models, are highest for Status 7 patients and lowest among Status 2 patients. Status 2 patients, however, do continue to receive implants through 10 years from listing on the UNOS waitlist (Fig 3g). The Cox model predicts this same pattern: as all other statuses stop receiving implants, Status 2 patients are continuing to increase their cumulative incidences (Fig 3h). The Fine-Gray model, instead, predicts a much higher gap in implant rates between Status 2 patients and all others (Fig 3i).

## Discussion

Patients listed at the lowest priority (UNOS Status 2) appear disadvantaged as they eventually experience higher cumulative incidences of death compared to patients at higher priorities, yet they are least likely to require implants.

We chose to stratify the data by UNOS Status to highlight the strengths and shortcomings of the current heart allocation policy. In doing so, we discovered that patients at the lowest priority level, UNOS Status 2 patients, are at a disadvantage with respect to allocation. Fig 3a depicts that the rate of death of Status 2 patients eventually surpasses those of patients at higher priorities. Fig 3d highlights the current UNOS Status prioritization for transplantation accurately. Implant events are shown to be least viable for Status 2 patients as they make up the group with the lowest probability of receiving one. Fig 3g shows that the rate of implantation up to Day 500 for Status 2 patients is much lower than those of other Statuses. The implantation and death incidences highlight how the prioritization of Status 2 patients proves to be one of the largest challenges facing this current policy during a patient's first few years on the waitlist.

Patients on the UNOS waitlist will have changes in condition and require reallocation to new Statuses. Our study uses patient UNOS Status only at the time of listing. However, the UNOS STAR files do not maintain historical changes in the variables we used (Table 3), thus prohibiting time-varying analyses. Our study also used the covariates found only by using the cause-specific hazards model; more-appropriate ways for identifying covariates using the sub-distribution hazards model are under development [18–20]. Regardless, the event rates calculated by our models pose extreme inconsistencies within current prioritization practices. One would expect the graphs of different events to follow a highest- to lowest-priority order. For example, Fig 3d displays this trend. Those with the highest probability of receiving a transplant are also the highest priority patients, and those with the lowest probability are the lowest priority. This graph remains as the only depiction of those policies while the graphs for the other two events, death (Fig 3a) and implant (Fig 3g), show that given initial waitlist priorities, there are unexpected patient outcomes with respect to priority.

The competing risks model posits a strength of predicting survival probabilities for patients with an upward bias as indicated by the Kaplan-Meier estimate. In our study, implantation acts as a competing event once patients are listed on the waitlist. MCSD clinical

recommendations [5] indicate that MCSDs are a BTT therapy for patients, which affects patient health trajectories and UNOS prioritization. Thus, our study excludes patients with MCSDs prior to listing for transplantation (i.e., patients with higher priority). However, MCSDs are affording long-term survival from heart failure [5], thus our study underestimates the disadvantages of Status 2 patients.

The US heart allocation policy has undergone some recent changes [4]. In contrast to other organ allocation policies, however, a prioritization model incorporating clinical characteristics has yet to be developed for the heart allocation policy. While different models have been proposed [21–23], a consensus on a prevailing model has yet to be reached [23]. When one model prevails, the information it requires must be collected by UNOS to properly assign waitlist priority. Recent policy changes address the excessive amount of Status 1A patients, the exorbitant exception requests, incorrect allocation of patients with MCSD based on prognosis, and geographical disparities as a consequence of the current policy [4]. Stratification by MCSD type does not address the disadvantages currently faced by Status 2 patients, who appear to have more profound reasons for not being candidates for MCSD implantation yet eventually die at a higher rate than higher-priority patients. The clinical characteristics of patients with end-stage heart failure should be accommodated as factors in the prioritization of the UNOS heart transplant waitlist.

## Acknowledgments

The authors are grateful to the academic editor and two anonymous reviewers for their comments to earlier versions of this manuscript. The data reported here have been supplied by the United Network for Organ Sharing as the contractor for the Organ Procurement and Transplantation Network. The interpretation and reporting of these data are the responsibility of the author(s) and in no way should be seen as an official policy of or interpretation by the OPTN or the US Government.

## Author Contributions

**Conceptualization:** Kevin B. Smith, Tseeye Odugba Potters, Gabriel Lopez Zenarosa.

**Data curation:** Kevin B. Smith, Tseeye Odugba Potters, Gabriel Lopez Zenarosa.

**Formal analysis:** Kevin B. Smith, Tseeye Odugba Potters, Gabriel Lopez Zenarosa.

**Funding acquisition:** Kevin B. Smith, Gabriel Lopez Zenarosa.

**Investigation:** Kevin B. Smith, Tseeye Odugba Potters, Gabriel Lopez Zenarosa.

**Methodology:** Kevin B. Smith, Tseeye Odugba Potters, Gabriel Lopez Zenarosa.

**Project administration:** Gabriel Lopez Zenarosa.

**Resources:** Gabriel Lopez Zenarosa.

**Software:** Kevin B. Smith, Tseeye Odugba Potters, Gabriel Lopez Zenarosa.

**Supervision:** Gabriel Lopez Zenarosa.

**Validation:** Gabriel Lopez Zenarosa.

**Visualization:** Kevin B. Smith, Gabriel Lopez Zenarosa.

**Writing – original draft:** Kevin B. Smith, Tseeye Odugba Potters, Gabriel Lopez Zenarosa.

**Writing – review & editing:** Kevin B. Smith, Gabriel Lopez Zenarosa.

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
