## [Decision Letter · Decision Letter 0]

7 Mar 2022

PONE-D-21-27074Pretransplant survival of patients with end-stage heart failure under competing risksPLOS ONE

Dear Dr. Zenarosa,

Thank you for submitting your manuscript to PLOS ONE. After careful consideration, we feel that it has merit but does not fully meet PLOS ONE’s publication criteria as it currently stands. Therefore, we invite you to submit a revised version of the manuscript that addresses the points raised during the review process.

We look forward to receiving your revised manuscript.

Kind regards,

Yoshihiro Fukumoto

Academic Editor

PLOS ONE

“This work was supported by funds provided by The University of North Carolina at

Charlotte, University Professional Internship Program, Levine Scholars Program, and

Honors College. Funding sources had no involvement in this study. The corresponding

author affirms that he has listed everyone who contributed significantly to the work.”

“This work was supported by funds provided by The University of North Carolina at Charlotte (GLZ, Faculty Research Grant, https://research.charlotte.edu/departments/center-research-excellence-cre/locating-funding/internal-funding-programs), University Professional Internship Program (KBS, https://career.charlotte.edu/upip), Levine Scholars Program (KBS, https://levinescholars.charlotte.edu), and Honors College (KBS, https://honorscollege.charlotte.edu). The funders had no role in study design, data collection and analysis, decision to publish, or preparation of the manuscript.”

Reviewers' comments:

Reviewer's Responses to Questions

**Comments to the Author**

1. Is the manuscript technically sound, and do the data support the conclusions?

Reviewer #1: Partly

Reviewer #2: Partly

2. Has the statistical analysis been performed appropriately and rigorously? 

Reviewer #1: I Don't Know

Reviewer #2: I Don't Know

3. Have the authors made all data underlying the findings in their manuscript fully available?

Reviewer #1: Yes

Reviewer #2: No

4. Is the manuscript presented in an intelligible fashion and written in standard English?

Reviewer #1: Yes

Reviewer #2: Yes

5. Review Comments to the Author

Reviewer #1: The authors reported the survival analyses of candidates for heart transplantation list under competing events of transplantation and MCSD implantation. The authors aimed to measure properly and estimate the survival probability of patients on the UNOS waitlist under competing risks to potentially highlight systematic biases and possible areas of improvement in organ allocation and patient status classification. However, this reviewer considers that this paper did not well meet the results for the objectives. This reviewer has several comments as described below.

Major comments.

1. The study found high mortality and low attainment of transplantation in Status 2 patients. The authors should indicate the predictors of mortality in Status 2 patients. One of them might be small left ventricular diameters, such as restrictive cardiomyopathy, who cannot fit to VAD.

2. This reviewer feels that it was to overstate to conclude that all status2 patients have high mortality. The authors should explain what clinical features of status 2 had high mortality.

3. It is important to point out that the mortality rate of status 2 is higher than that of status 1 when the observation period is prolonged. It may indicate that the current organ distribution system is acceptable up to 300 days or 1002 days. The authors should add that point.

4. As the authors indicated, the results did not reflect the changes in status during the study period, which may lead to problems with the accuracy of the results. This was a big limitation.

5. Dissociation of the results between the cause-specific Hazards Model and the Subdistribution Hazards Model was difficult to understand, which should be explained more carefully.

6. In the Discussion section, the author described that Status2 patients had a higher rate of death than patients at higher priorities in Fig 2a. Was this Fig 3a?

Reviewer #2: Describe heart-allocation policy and details about status /priority IA, 1B, 2 and 7.

Figure 2 should present cumulative Incidence of Patient Survival event for each category of UNOS.

Is the waiting list kept updated frequently as patient health conditions evolve? Did the priorities of patients analyzed in the study change accordingly?

As shown in Fig 3a, during the first 500 days, the death probability in status 2 group is the lowest. Afterwards, the death probability rises up and exceeds d status 1B and 7 starting from day 1500.

Label for each group should be distinguished. As shown in Fig.3 d, status 1b and status 2 has continuously received implantations through the entire 10 years, not as the authors stated “only Status 2 patients are receiving transplantation after 500 days and before 2000 days “.

Provide waiting time information for transplantation and implantation in each group/status. The plot between waiting time for transplantation/ implantation and death should be provided, which are more valuable and informative.

Page 8 lines 155-156 the authors wrote “ Fig 2a depicts that Status 2 patients have a higher rate of death 155 than patients at higher priorities.” However, Fig.2a did not provide any information about priorities.

On page 9 Lin 185-186 Sentence “Long proposed is the idea that a score for patients with end-stage heart failure should be developed and utilized to judge priority..” What is a score? When is it collected?

There are many grammar and typo errors. And English proof is needed. For example, on Page3 line 45 “ aged less than 16 years” ; On page 9 Lin 185-186 Sentence “Long proposed is the idea that a score for patients with end-stage heart failure should be developed and utilized to judge priority.. “

Full name for Abbreviation should be shown when it appears the first time. For example, UNOS.

6. PLOS authors have the option to publish the peer review history of their article (what does this mean?). If published, this will include your full peer review and any attached files.

Reviewer #1: No

Reviewer #2: No

---

## [Author Response · Author response to Decision Letter 0]

25 May 2022

We are grateful to the publisher, academic editor, and two anonymous reviewers for their comments to improve our manuscript. Summarized in the following sections are the itemized comments accompanied by our corresponding responses and/or updates to the manuscript.

Note: References to manuscript Line numbers are those of the change-tracked (marked-up) version.

Academic Editor’s Comments and Authors’ Responses

Response: We use the PLOS ONE LaTeX template (https://journals.plos.org/plosone/s/latex), and:

• We revised our manuscript to conform to the formatting guidelines, including correcting the author notes including affiliations, addresses, and corresponding authorship, as well as manually adjusting the default LaTeX template line-spacing for the title page.

• We corrected the citations at the ends of sentences to appear before the punctuations.

• We renamed the figure files as Fig1.tiff, Fig2.tiff, and Fig3.tiff.

• We corrected the caption and legend of Table 3.

• We adjusted for the LaTeX template automation to force Table 3 to appear within the manuscript rather than after the References. However, the LaTeX template still places Tables 1–3 on the pages following their first citations because they do not fit within the page under the paragraph in which they were first cited.

“This work was supported by funds provided by The University of North Carolina at

Charlotte, University Professional Internship Program, Levine Scholars Program, and

Honors College. Funding sources had no involvement in this study. The corresponding

author affirms that he has listed everyone who contributed significantly to the work.”

“This work was supported by funds provided by The University of North Carolina at Charlotte (GLZ, Faculty Research Grant, https://research.charlotte.edu/departments/center-research-excellence-cre/locating-funding/internal-funding-programs), University Professional Internship Program (KBS, https://career.charlotte.edu/upip), Levine Scholars Program (KBS, https://levinescholars.charlotte.edu), and Honors College (KBS, https://honorscollege.charlotte.edu). The funders had no role in study design, data collection and analysis, decision to publish, or preparation of the manuscript.”

Response: We removed the funding information from the Acknowledgements section and updated its contents as follows:

The authors are grateful to the academic editor and two anonymous reviewers for their comments to earlier versions of this manuscript. The data reported here have been supplied by the United Network for Organ Sharing as the contractor for the Organ Procurement and Transplantation Network. The interpretation and reporting of these data are the responsibility of the author(s) and in no way should be seen as an official policy of or interpretation by the OPTN or the U.S. Government.

We accept the funding statement as suggested, which is repeated here:

This work was supported by funds provided by The University of North Carolina at Charlotte (GLZ, Faculty Research Grant, https://research.charlotte.edu/departments/center-research-excellence-cre/locating-funding/internal-funding-programs), University Professional Internship Program (KBS, https://career.charlotte.edu/upip), Levine Scholars Program (KBS, https://levinescholars.charlotte.edu), and Honors College (KBS, https://honorscollege.charlotte.edu). The funders had no role in study design, data collection and analysis, decision to publish, or preparation of the manuscript.

Response: The UNOS Data Use Agreement signed by the UNC Charlotte liaison prohibits us from releasing the dataset without approval, which is quoted here: “You will neither release nor permit other to release the Data to any person (including media and subcontractors) except with the written approval of UNOS.” 

As a result, we would like to update our Data Availability Statement (as similarly stated in another PLOS ONE article using the same transplantation dataset, https://dx.doi.org/10.1371%2Fjournal.pone.0247789):

Data cannot be shared publicly as it is owned by the United Network for Organ Sharing. We do not have permission to distribute the data, however, the data may be requested from the Organ Procurement and Transplantation Network. To obtain the data, a Data Use Agreement must be signed and approved by OPTN. Please refer the following URL: https://optn.transplant.hrsa.gov/data/request-data/.

Response: Kindly see our response to the preceding comment, which also applies to this comment.

Response: We moved our full ethics statement from the Patient Selection subsection to a new Ethics subsection (Lines 35–42) at the beginning of the Methods section.

Reviewer #1’s Comments and Authors’ Responses

The authors reported the survival analyses of candidates for heart transplantation list under competing events of transplantation and MCSD implantation. The authors aimed to measure properly and estimate the survival probability of patients on the UNOS waitlist under competing risks to potentially highlight systematic biases and possible areas of improvement in organ allocation and patient status classification. However, this reviewer considers that this paper did not well meet the results for the objectives. This reviewer has several comments as described below.

Major comments.

1. The study found high mortality and low attainment of transplantation in Status 2 patients. The authors should indicate the predictors of mortality in Status 2 patients. One of them might be small left ventricular diameters, such as restrictive cardiomyopathy, who cannot fit to VAD.

Response: We include all at-listing variables considered in our models in Table 3. We state in Lines 64–70 that we considered all at-listing variables available from the data set and perform a mixed-selection method to derive the covariates for our models. Clinical characteristics, such as “small left ventricular diameters,” were not available in the dataset. Nevertheless, patients that may have been diagnosed with restrictive myopathy before or at listing would be contained within our primary diagnosis categorical variable.

2. This reviewer feels that it was to overstate to conclude that all status2 patients have high mortality. The authors should explain what clinical features of status 2 had high mortality.

Response: We corrected our previous overstatement on Lines 162–163:

Fig 3a depicts that the rate of death of Status 2 patients eventually surpasses those of patients at higher priorities.

The death hazard ratios for the variables we analyzed are tabulated in Table 3.

3. It is important to point out that the mortality rate of status 2 is higher than that of status 1 when the observation period is prolonged. It may indicate that the current organ distribution system is acceptable up to 300 days or 1002 days. The authors should add that point.

Response: We respectfully disagree: An equitable prioritization and allocation policy must attempt to maintain mortality curves ordered in accordance with priority, especially under competing risks: Status 2 patients have less alternatives. Throughout the 10-year study period, Status 2 patients have lower incidence of transplants (as imposed by their low priority), and the analyses revealed they also have lower incidence of implants.

4. As the authors indicated, the results did not reflect the changes in status during the study period, which may lead to problems with the accuracy of the results. This was a big limitation.

Response: We agree; however, the available data does not allow for time-varying analyses, which we clarified in the new sentence added on Lines 174–176:

However, the UNOS STAR files do not maintain historical changes in the variables we used (Table 3), thus prohibiting time-varying analyses.

Nonetheless, we believe we modeled the problem as appropriately as contemporary methodology allows.

5. Dissociation of the results between the cause-specific Hazards Model and the Subdistribution Hazards Model was difficult to understand, which should be explained more carefully.

Response: We revised the Predictors of Survival subsection of the Results section to clarify this dissociation in Lines 104–109:

We note that the cause-specific hazard ratio (csHR) represents the rate of the event of interest in those patients that are event-free; thus, csHR provides the estimated etiological effects of the variables. In contrast, the subdistribution hazard ratio (sdHR) provides the prognostic effects of the variables.

We also clarified the respective effects (in the context of an example) in the paragraph containing Lines 123–126:

… we present csHRs and sdHRs to provide decision-makers the complete estimated etiological and prognostic effects, respectively, of the variables in our multivariate analyses.

Additionally, we separated the discussion of exercising care in the interpretation of the subdistribution Hazards Model into its own paragraph:

clarify this dissociation on Lines 104–109:

The interpretation of the sdHR requires some care, however. We can assume that if a variable increases the subdistribution hazard, it will also increase the incidence of the event of interest, but we cannot conclude that these two are in the same magnitude. Thus, using the value of a covariate's sdHR only approximately describes the effect of that variable on the incidence of the event of interest.

6. In the Discussion section, the author described that Status2 patients had a higher rate of death than patients at higher priorities in Fig 2a. Was this Fig 3a?

Response: We corrected the figure reference to Fig 3a.

Reviewer #2’s Comments and Authors’ Responses

1. Describe heart-allocation policy and details about status /priority IA, 1B, 2 and 7.

Response: We describe study-relevant information on heart allocation, including the citation to the full policy, as well as UNOS Statuses in Lines 51–54.

2. Figure 2 should present cumulative Incidence of Patient Survival event for each category of UNOS.

Response: Cumulative incidences of patient survival for all UNOS Statuses are presented in Fig 3a.

3. Is the waiting list kept updated frequently as patient health conditions evolve? Did the priorities of patients analyzed in the study change accordingly? As shown in Fig 3a, during the first 500 days, the death probability in status 2 group is the lowest. Afterwards, the death probability rises up and exceeds d status 1B and 7 starting from day 1500.

Response: Kindly refer to our responses to Reviewer 1’s Comments 4 and 5 above, which apply here.

4. Label for each group should be distinguished. As shown in Fig.3 d, status 1b and status 2 has continuously received implantations through the entire 10 years, not as the authors stated “only Status 2 patients are receiving transplantation after 500 days and before 2000 days “.

Response: We distinguish UNOS Status for all panels of Fig 3 using the legend found in the upper left corner. Fig 3d displays the cumulative incidence of the transplantation event, not the implantation event. Fig 3d shows that the Status 1A and Status 1B cumulative incidence of transplantation are constant (i.e., flat curves) after 500 and 2000 days, respectively; this indicates that no transplants have since been accumulated for those statuses.

5. Provide waiting time information for transplantation and implantation in each group/status. The plot between waiting time for transplantation/ implantation and death should be provided, which are more valuable and informative.

Response: Cumulative incidence curves incorporate waiting time by design; the step increases in Figs 3a, 3d, and 3g depict on the y-axis the incremental proportion of patients who experienced the event of interest (i.e., after the last step increase), and their waiting time is equal to the Days to Event on the x-axis when the step increase occurred.

6. Page 8 lines 155-156 the authors wrote “ Fig 2a depicts that Status 2 patients have a higher rate of death 155 than patients at higher priorities.” However, Fig.2a did not provide any information about priorities.

Response: We corrected the figure reference to Fig 3a.

7. On page 9 Lin 185-186 Sentence “Long proposed is the idea that a score for patients with end-stage heart failure should be developed and utilized to judge priority..” What is a score? When is it collected?

Response: We clarified the sentence with the following replacement on Lines 199–202:

In contrast to other organ allocation policies, however, a prioritization model incorporating clinical characteristics has yet to be developed for the heart allocation policy. While different models have been proposed, a consensus on a prevailing model has yet to be reached.

8. There are many grammar and typo errors. And English proof is needed. For example, on Page3 line 45 “ aged less than 16 years” ; On page 9 Lin 185-186 Sentence “Long proposed is the idea that a score for patients with end-stage heart failure should be developed and utilized to judge priority.. “

Response: We respectfully disagree: The two examples given are grammatically correct. We proofread the manuscript again as suggested.

9. Full name for Abbreviation should be shown when it appears the first time. For example, UNOS.

Response: We corrected this on Lines 23–24 and Lines 29–30.

---

## [Decision Letter · Decision Letter 1]

3 Aug 2022

Pretransplant survival of patients with end-stage heart failure under competing risks

PONE-D-21-27074R1

Dear Dr. Zenarosa,

We’re pleased to inform you that your manuscript has been judged scientifically suitable for publication and will be formally accepted for publication once it meets all outstanding technical requirements.

Kind regards,

Yoshihiro Fukumoto

Academic Editor

PLOS ONE

Additional Editor Comments (optional):

Reviewers' comments:

Reviewer's Responses to Questions

**Comments to the Author**

1. If the authors have adequately addressed your comments raised in a previous round of review and you feel that this manuscript is now acceptable for publication, you may indicate that here to bypass the “Comments to the Author” section, enter your conflict of interest statement in the “Confidential to Editor” section, and submit your "Accept" recommendation.

Reviewer #1: All comments have been addressed

2. Is the manuscript technically sound, and do the data support the conclusions?

Reviewer #1: Partly

3. Has the statistical analysis been performed appropriately and rigorously? 

Reviewer #1: I Don't Know

4. Have the authors made all data underlying the findings in their manuscript fully available?

Reviewer #1: Yes

5. Is the manuscript presented in an intelligible fashion and written in standard English?

Reviewer #1: Yes

6. Review Comments to the Author

Reviewer #1: This reviewer has no further comment. The authors indicated to reconsider the classification of priorities in heart transplantation. It is a future challenge to determine what clinical characteristics of Status 2 patients are predictors of higher priority for heart transplantation. This reviewer hopes that this manuscript will serve as a starting point.

7. PLOS authors have the option to publish the peer review history of their article (what does this mean?). If published, this will include your full peer review and any attached files.

Reviewer #1: No

---

## [Editor Report · Acceptance letter]

5 Aug 2022

PONE-D-21-27074R1 

Pretransplant survival of patients with end-stage heart failure under competing risks 

Dear Dr. Zenarosa:

I'm pleased to inform you that your manuscript has been deemed suitable for publication in PLOS ONE. Congratulations! Your manuscript is now with our production department. 

Kind regards, 

on behalf of

Dr. Yoshihiro Fukumoto 

Academic Editor

PLOS ONE